# Association Between Intensity of Physical Activity in Pregnancy and Gestational Diabetes in a Multi-Ethnic Population: Results from the PROMOTE Cohort Study

**DOI:** 10.3390/nu17223500

**Published:** 2025-11-07

**Authors:** Ania (Lucewicz) Samarawickrama, James Elhindi, Yoon Ji Jina Rhou, Sarah J. Melov, Vicki Flood, Justin McNab, Mark McLean, Ngai Wah Cheung, Ben J. Smith, Tim Usherwood, Dharmintra Pasupathy

**Affiliations:** 1Reproduction and Perinatal Centre, Faculty of Medicine and Health, The University of Sydney, Sydney 2145, Australia; 2General Practice Clinical School, Faculty of Medicine and Health, The University of Sydney, Sydney 2006, Australia; 3Specialty of Medicine, Faculty of Medicine and Health, The University of Sydney, Sydney 2006, Australia; 4Department of Diabetes & Endocrinology, Westmead Hospital, Sydney 2145, Australia; 5Westmead Institute for Maternal and Fetal Medicine, Westmead 2145, Australia; 6University Centre for Rural Health, Northern Rivers Clinical School, The University of Sydney, Sydney 2480, Australia; 7Sydney School of Public Health, The University of Sydney, Sydney 2006, Australia; 8Westmead Institute for Medical Research, Westmead 2145, Australia; 9George Institute for Global Health, Sydney 2000, Australia; 10Specialty of Obstetrics, Gynaecology and Neonatology, Westmead Clinical School, Faculty of Medicine and Health, The University of Sydney, Sydney 2145, Australia

**Keywords:** epidemiology, perinatal, birth cohort study, obstetrics, gestational diabetes, physical activity, obesity, lifestyle

## Abstract

**Introduction**: The demographic shift amongst pregnant women, including older age and increasing obesity, has resulted in an increased risk of cardiometabolic complications during pregnancy, particularly gestational diabetes. This paper presents physical activity and gestational diabetes data in a multi-ethnic urban Australian population. **Methods and analysis**: The PROMOTE cohort study is an ongoing prospective pregnancy cohort study recruiting pregnant participants < 16 weeks gestation at a large urban public teaching hospital with high social and cultural diversity in Sydney, Australia. Participants are surveyed about their physical activity levels, dietary quality, emotional wellbeing and socio-demographic status using validated tools. Participants are consented for use of routinely collected clinical and social data, including medical conditions, body mass index (BMI), blood pressure (BP) and glycaemia. Follow-up is from routinely collected data. **Results**: A total of 459 participants were recruited between February 2022 and February 2024. Physical activity levels at recruitment were sufficiently active, low active and inactive in 39%, 45% and 16% of participants. Participation in moderate or vigorous physical activity was reported in 19% and 16% of participants, respectively. Participation in vigorous physical activity occurred in 10% of those with GDM vs. 17% of those without GDM (*p* = 0.11). Participation in any moderate/vigorous physical activity was reported in 20% of those with GDM vs. 30% of those without GDM (*p* = 0.058). Compared to inactive behaviour, the unadjusted odds ratio of developing GDM amongst those participating in any moderate/vigorous physical activity was 0.58 (95% CI 0.33–0.97), *p* = 0.045. Participation in any moderate/vigorous physical activity showed an association with lower oral glucose tolerance test levels at 1 h (7.49 vs. 8.17 mmol/L, *p* = 0.002). Participation in any vigorous activity was associated with lower oral glucose tolerance test levels at 1 h (7.25 vs. 8.11, *p* = <0.001). **Conclusions**: Uptake of existing physical activity recommendations is low. Gestational diabetes risk showed a trend toward variation by intensity of physical activity, with a trend toward greater intensity being associated with a possible lower rate of gestational diabetes.

## 1. Introduction

### 1.1. Background/Rationale

Physical activity is widely recommended during pregnancy [1,2]. Observational studies have reported a range of positive associations [3], including optimal gestational weight gain [4], reduced risk of gestational diabetes (GDM) [5,6], improved perinatal mental health outcomes [7,8] and reduced musculoskeletal pain [9].

However, the type, duration and intensity of physical activity associated with clinical benefits during pregnancy, particularly GDM, remains unclear. Approaches to measurement of physical activity in pregnancy are highly varied. Despite positive observational evidence, systematic reviews of interventional trials of physical activity during pregnancy have reported mixed results [10,11,12]. A wide range of reviews have been published, seeking to make sense of the diversity in measurement and reporting of physical activity, variation in clinical endpoints and timing and target population of trials [13,14,15,16,17,18]. Although more recent reviews emphasising more recent randomised controlled trials and systematic reviews trend towards positive effects [19], questions around type, duration and intensity of physical activity in pregnancy remain.

Despite well documented evidence of ethnic variation in gestational diabetes risk, concerns have been raised around the lack of trial representation for non-white women in the evidence base examining the relationship between lifestyle exposures and gestational diabetes risk [20]. There is a need to diversify the evidence base to improve its relevance.

Finally, despite well documented guidance recommending physical activity in pregnancy, uptake of recommendations remains low [21]. Understanding predictors and sociodemographic associations with physical activity is crucial to the design of targeted interventions in a multi-ethnic population.

### 1.2. Aim

Here, we present results from the PROMOTE Cohort Study, reporting the association between participation in physical activity in pregnancy, of varying intensities and durations, and gestational diabetes. Furthermore, we examined associations between levels of physical activity and (i) results of oral glucose tolerance tests (OGTT) and (ii) clinical and social characteristics.

## 2. Methods

### 2.1. Overall Study Design and Population

The PROMOTE Cohort Study is a prospective pregnancy cohort study conducted in a socioeconomically and ethnically diverse population in western Sydney, Australia. The overall design and rationale of the PROMOTE Cohort Study has been described elsewhere [22]. Briefly, women seeking antenatal care at a large, urban, state-funded hospital are offered recruitment in early pregnancy (<16 weeks gestation). Participants are consented for use of routinely collected clinical data for research purposes, which includes a wide array of clinical history, baseline and longitudinal measurements and pathology results. Additionally, participants are requested to complete surveys about their diet, physical activity levels, sociodemographic circumstances and mental health and optionally offered biobanking of maternal and cord blood samples [22]. In keeping with the published protocol, data from participants recruited between February 2022 and February 2024 were analysed. The study is entirely observational and no lifestyle intervention or counselling is performed. Usual care is provided by clinical staff and study staff do not provide clinical services.

### 2.2. Data Collection for Baseline Characteristics

Baseline characteristics of the study population were extracted from routine clinical records and include visit characteristics, participant demographics and obstetric and medical history. Clinical measurements, including height, weight, body mass index (BMI) and blood pressure were extracted from routine clinical records and reflect their performance during routine clinical care. Mental health screening was performed in routine clinical practice by way of the Edinburgh Perinatal Depression Scale (EPDS) [23], and participants were also requested to complete a Depression Anxiety and Stress Scale (DASS21) [24]. Additionally, participants were asked about additional sociodemographic characteristics via a bespoke questionnaire (Appendix A).

### 2.3. Data Collection for Physical Activity

Physical activity was measured using the Active Australia Survey (AAS), delivered at recruitment. This was prior to 16 weeks’ gestation and reflects physical activity behaviours in the past week [25]. The AAS consists of eight questions designed to measure participation in leisure-time physical activity and five statements to assess knowledge about health benefits of physical activity. The survey is included in Appendix A.

Physical activity was categorised as sufficiently active, low active and inactive. The reporting guideline for the tool categorises physical activity into the following categories: sufficient, insufficient or sedentary [25]. However, since the publication of the reporting guideline, terminology in the field has progressed and ‘sedentary’ is now understood to mean prolonged immobility [26]. As the AAQ is a leisure time physical activity measure, those doing little to no reported activity may not meet the new definition of sedentary. To avoid misinterpretation, category terminology was amended and reported as sufficiently active, low active and inactive.

Physical activity category was treated as ‘sufficiently active’ if the participant reported an accumulation of a sufficient amount of activity over a week or the accumulation of a sufficient amount of activity by participation in a sufficient number of sessions over a week. This reflected 30 min on at least 5 days of the week or a total of at least 150 min per week. Physical activity was regarded as ‘inactive’ if the participant had no leisure time physical activity or walking during the previous week. Activity that was neither sufficiently active nor inactive was classified as low active.

Additionally, granular subcategorization of physical activity behaviours was performed. Subcategorization by duration, frequency and intensity was performed by disaggregating questions about walking time, moderate physical activity time and vigorous physical activity time. Moderate and vigorous were defined using self-perceived and self-reported measures as per the AAS reporting handbook, where moderate physical activity increases heart rate but does not necessarily make a person puff or pant and includes activities such as walking, golf, gentle swimming and social tennis. Vigorous activities are described as those that make a person breathe harder or puff and pant and include activities such as jogging, cycling, aerobics and competitive tennis. Derived data elements are reported as moderate physical activity time and vigorous physical activity time and reported as a binary variable (yes/no) [25].

Participant understanding of public health messaging around physical activity levels was assessed by questions 9(a)–(e) of the Active Australia Survey. These questions assess participant understanding of five key public health messages around physical activity. Messaging agreement indicates awareness of public health messaging and is assessed using standard protocol outlined in the reporting manual for the survey [25].

### 2.4. Data Collection for Gestational Diabetes

Data about the incidence of gestational diabetes were extracted from routine clinical records. At our centre, universal screening for GDM is standard practice. All women not known to have diabetes mellitus (DM) or GDM are routinely invited to have a standard 2 h 75 g oral glucose tolerance test (OGTT) at 24–28 weeks’ gestation as part of their routine care, with some women also being offered an early OGTT on the basis of risk factors including age, BMI, ethnicity and previous history of GDM [27]. GDM was treated as a binary outcome using the Australasian Diabetes in Pregnancy Society (ADIPS) criteria [27] for the diagnosis of GDM. A diagnosis of GDM is made if one or more of the following glucose levels are elevated: fasting ≥ 5.1 mmol/L; 1 h glucose ≥ 10.0 mmol/L; 2 h glucose ≥ 8.5 mmol/L. Additionally, absolute venous glucose results at 0, 1 and 2 h were also extracted from the clinical record. Those without GDM or OGTT results available were excluded from analysis.

### 2.5. Study Size

We report here the analysis of the first two years of recruitment into the PROMOTE Cohort Study and include GDM outcomes of those who had completed an OGTT at the time of analysis. Recruitment is ongoing. As per the published protocol [22], a pragmatic approach to early analysis was adopted, using the first two years as the point at which analysis should be performed, with the goal that data from the first two years will be used to inform future sample size calculation. An a priori sample size calculation was not performed due to limitations in available empirical data about the distribution of lifestyle risk factors in our population, including sedentary and inactive behaviours.

### 2.6. Statistical Methods

Statistical analysis was conducted in R Studio Version 4. Hypotheses were conducted at a family-wise significance level of 0.05 with a two-sided alternative. Appropriate Bonferroni corrections were made to the significance level of individual hypotheses. Summary statistics describe the characteristics of the cohort. Descriptive statistics are presented on the relationship between outcomes of interest and baseline risk factors. Continuous variables are summarised as mean ± standard deviation if normally distributed and median and interquartile range if not. Categorical variables are presented as frequency (%) in relevant categories. For multivariable analyses, variables adjusted for include age, BMI, ethnicity and multiparity. Missing data was managed through deletion, and number of participants with an available data point are reported.

### 2.7. Ethics

The study has been approved by the Western Sydney Local Health District Human Research Ethics Committee (2021/ETH00287, 7 June 2021).

## 3. Results

### 3.1. Participants

In the period March 2022–end of February 2024, 582 potential participants were approached for recruitment and 507 (87%) agreed to participate in the study. Of these, 7 participants withdrew from the study, 9 were expecting twins, 6 could not be matched to an electronic medical record and 26 women had pre-existing diabetes and were excluded from this analysis, leaving a total of 459 (78% of those approached, 90% of those initially agreeable to recruitment) participants for analysis. See Figure 1 for the CONSORT diagram. See also Appendix A for a visual representation of key findings.

### 3.2. Baseline and Sociodemographic Characteristics

The mean age of the 459 participants was 32.48 years (range: 21–43). Participant characteristics are summarised in Table 1. Mean BMI was 25.3 kg/m^2^ (range: 16.8–50.9). Mean gestation at first assessment was 6.2 weeks (range 5–13 weeks), and mean gestational age at first antenatal hospital (booking) visit was 13 weeks (range: 5–25). There were 140 (30.5%) participants who were primiparous. A history of previous GDM was noted in 63 participants (18% amongst those who had previously been pregnant).

With respect to country of birth, 143 (31%) participants were born in Australia. Of the 459 total participants, 316 (69%) participants were born elsewhere, with India (*n* = 112; 24.5% of total cohort), Pakistan (*n* = 22; 4.8%), Nepal (*n* = 22; 4.8%), China (*n* = 26; 5.7%) and Afghanistan (*n* = 18; 3.9%) being the most common countries of birth. The most common self-reported ethnicity was South Asian (*n* = 172; 37%), White (*n* = 90; 20%), Middle Eastern (*n* = 81; 18%) and South-East Asian (*n* = 77; 17%). Three participants identified as Aboriginal or Torres Strait Islander (0.65%).

Socio-Economic Indexes for Areas scores were available for 351 participants and are summarised in Appendix A. Household income in Australian dollars was available for 416 participants and reported in categories, with the following number of participants in each bracket: <AUD 50,000 (*n* = 23; 5%), AUD 50–100,000 (*n* = 96; 21%), AUD 100–200,000 (*n* = 153; 33%), > AUD 200,000 (*n* = 75; 16.5%). In addition, 107 (23.5%) participants selected ‘Don’t know/Prefer not to say’. Education levels were as follows: incomplete high school (*n* = 10; 2%), completed high school (*n* = 38; 8.3%), completion of high school plus a post school qualification such as a certificate/diploma (*n* = 82; 18%) and completion of high school plus a university qualification (*n* = 317; 69%). Nine (1.9%) participants reported smoking and three (0.6%) reported illicit drug use. No participants reported alcohol use.

### 3.3. Physical Activity Length, Duration and Intensity

Category of physical activity was reported as follows and sumamrised in Table 2: sufficiently active (*n* = 179; 39%), low active (*n* = 206; 45%) and inactive (*n* = 73; 16%). With respect to intensity of physical activity, the following proportions reported any walking (*n* = 430, 94%), moderate (*n* = 85, 19%) and vigorous activity (*n* = 74, 16%). Overall, participation in any moderate or vigorous physical activity was reported by 25% of participants (*n* = 133).

### 3.4. Characteristics Associated with Physical Activity Behaviours

Characteristics associated with physical activity behaviours are reported in Appendix A. The only sociodemographic characteristics associated with participation in moderate and vigorous physical activities was ethnicity.

There was ethnic variation in intensity of physical activity, with any moderate or vigorous physical activity level reported as follows: 28/86 (33%) amongst those identifying as Middle Eastern, 32/176 (18%) amongst those identifying as South Asian, 23/84 (27%) amongst those identifying as South East Asian, 35/95 (37%) amongst those identifying as White and 15/44 (34%) amongst those identifying with Other ethnic groups.

There was no statistically significant association detected between intensity of physical activity and the following: maternal age, parity, education level, DASS21, EPDS, history of GDM and BLISS score in this cohort.

### 3.5. Physical Activity Behaviours Associated with Gestational Diabetes

Of the 459 participants, 416 (91.2%) had available GDM data at the time of analysis. Of these, 104 (25%) had GDM. Characteristics associated with gestational diabetes are summarised in Appendix A.

GDM status varied by intensity of physical activity reported and is summarised in Table 3. There was a trend towards lower levels of participation in vigorous physical activity amongst those with GDM compared to those without; these were 10% amongst those with GDM vs. 17% amongst those without GDM (*p* = 0.097). Participation in any form of moderate or vigorous activity was 20% amongst those with GDM vs. 30% amongst those without GDM (*p* = 0.058).

The GDM rate amongst those who participated in physical activity was as follows: any walking (*n* = 89/367, 24%), any moderate physical activity (*n* = 15/75, 20%), any vigorous physical activity (*n* = 10/63, 16%) and any moderate or vigorous activity together (*n* = 21/116, 18%).

Compared to those with inactive behaviour, the unadjusted odds ratio for developing GDM amongst those participating in any moderate physical activity was 0.71 (95% CI 0.37–1.28, *p* = 0.271); for those participating in any vigorous activity, it was 0.52 (95% CI 0.24–1.02, *p* = 0.073); and for those participating in any moderate or vigorous activity, it was 0.58 (95% CI 0.33–0.97, *p* = 0.045). When adjusted for age, BMI, ethnicity and multiparity, these odds ratios were 0.70 (95% CI 0.36–1.30, *p* = 0.275) for moderate, 0.59 (95% CI 0.27–1.20, *p* = 0.165) for vigorous and 0.59 (0.33–1.01, *p* = 0.061) for any moderate or vigorous activity (Table 4).

### 3.6. Physical Activity Behaviours and OGTT Results

Oral glucose tolerance test (OGTT) data was available for 379 (82.5%) participants. Results are summarised in Table 5. There was no statistically significant association detected between physical activity categories (inactive, low active and sufficiently active) and glucose levels, nor was there a significant association between participation in any walking and glucose levels. While fasting glucose levels did not vary by physical activity, participation in vigorous physical activity showed an association with lower glucose levels at 1 h (7.25 vs. 8.11 mmol/L, *p* < 0.001). Furthermore, when taken as a group, those that participated in any moderate or any vigorous physical activity also demonstrated lower glucose levels at 1 h (7.49 vs. 8.17 mmol/L, *p* = 0.002) (Table 5).

## 4. Discussion

The PROMOTE Cohort Study is the first pregnancy cohort study to be established in the highly ethnically and socioeconomically diverse setting of western Sydney, Australia. This setting enabled recruitment of previously under-represented cohorts, particularly those with migration history and heritage. Trial equity has been identified as a need in examination of lifestyle risk factors and GDM [20]. This study seeks to address the lack of physical activity data collected in multi-ethnic populations to better define these relationships.

Australian data around physical activity during pregnancy is limited. Last available AIHW data from the 2011–12 period suggests that two-thirds of Australian women did not participate in at least 150 min of physical activity per week and that only 3 in 10 women met physical activity recommendations during pregnancy [21]. Data suggests similar levels of inactivity postpartum [28]. Our data is consistent with these findings, observing that only 39% of surveyed women met physical activity recommendations during early pregnancy. In the general population, data from 2022 suggests that 41% of Australian women aged between 18 and 65 were insufficiently active [29], but national data around variation by demographic subgroup is limited.

Our study suggests a trend towards a possible association between increasing intensity of physical activity and reduction in GDM risk. Current Australian guidance is based on a relatively uniform set of recommendations for women but as yet does not tailor recommendations to pre-existing and varied cardiometabolic risk [30]. It is possible that appropriate recommendations for physical activity in pregnancy may require individualisation based on pre-existing cardiometabolic risk profile, such as BMI or body composition, ethnicity, co-existing conditions such as polycystic ovarian syndrome, age, etc.

Most published lifestyle interventions aiming to modify physical activity or other lifestyle factors in pregnancy in order to ameliorate the risk of GDM focus on delivering individual-level interventions (such as clinical consultation and information about physical activity) within a clinical setting [31]. Other interventions, such as individual health coaching, have focussed on reporting impact on health behaviours rather than clinical outcomes like gestational diabetes or birth outcomes [32]. There remains significant scope to widen the duration and design of interventions, including further co-design and trial of direct-to-consumer or digital interventions [33], place-based interventions, such as partnership with existing opportunities in the urban environment, social interventions, culturally adapted interventions and/or whole-family interventions.

In the general population, there is increasing acknowledgement that individual interventions may be limited where contextual factors remain unaddressed. It is increasingly being acknowledged that synergistic multi-level interventions spanning public health, policy, regulation, health promotion, urban design and food security, together with clinical interventions, may be needed to make meaningful change for a multi-factorial end-point like physical activity [34]. Such ‘socio-ecological’ approaches to health acknowledge wider determinants of health behaviours and suggest such factors warrant inclusion in the design and evaluation of clinical interventions.

Limitations of our study include the need to assess lifestyle exposures collectively, in particular dietary data alongside physical activity data. This analysis is planned. Dietary data is a crucial component of cardiometabolic risk, interacting with physical activity, BMI, gestational weight gain and other risk factors [35]. Furthermore, combined interventions have been most commonly trialled due to synergistic effects of diet and physical activity together [36]. Analysis of dietary patterns, alone and in combination, is planned.

Furthermore, the reliability of self-reported physical activity measures has also been queried, and evolving methods such as patient-generated data and accelerometry may offer potential alternatives [37,38,39,40]. Some of these digital ways of measuring physical activity, including the use of smart phones with in-built accelerometers have not only been used to measure physical activity but have been coupled with digital interventions [41,42]. Similarly, incorporating baseline measures of fitness and/or body composition may complement insights from measures of physical activity levels and clarify the relationship between physical activity and BMI.

Our study is also limited by the pragmatic approach to sample size. We present here the analysis of the first two years of recruitment. Empirical data gathered, particularly on the distribution of exposures in our population, will be used to inform more robust power and sample size calculations to inform future analyses. Recruitment is ongoing and this analysis is planned.

Another limitation of our study is the single-timepoint analysis. Longitudinal measurements of lifestyle exposures such as changes in physical activity, as well as baseline data around fitness and body composition, would likely complement the existing analysis. Additionally, further data around the type of physical activity (e.g., swimming, jogging, cycling, etc.) would be helpful to inform the design of interventions. Our study is also limited to a single area of Australia and would be strengthened by participants from a variety of settings.

Finally, we also acknowledge that better understanding of the relationship between intensity of physical activity and glucose levels at fasting, 1 h and 2 h intervals is key, and this analysis is planned.

Nevertheless, this study provides insight into physical activity during pregnancy in a multi-ethnic population.

## 5. Conclusions

Uptake of physical activity was low in our sample. This was true across all demographic groups, with some variation by financial status and ethnicity. Further work is necessary to understand and better define the nature of this association, as well as incorporate these insights into the design of future interventions. Participation in any moderate or vigorous physical activity showed an association with a trend towards a possible reduction in GDM risk and in OGTT glucose level at 1 h.

## Figures and Tables

**Figure 1 nutrients-17-03500-f001:**
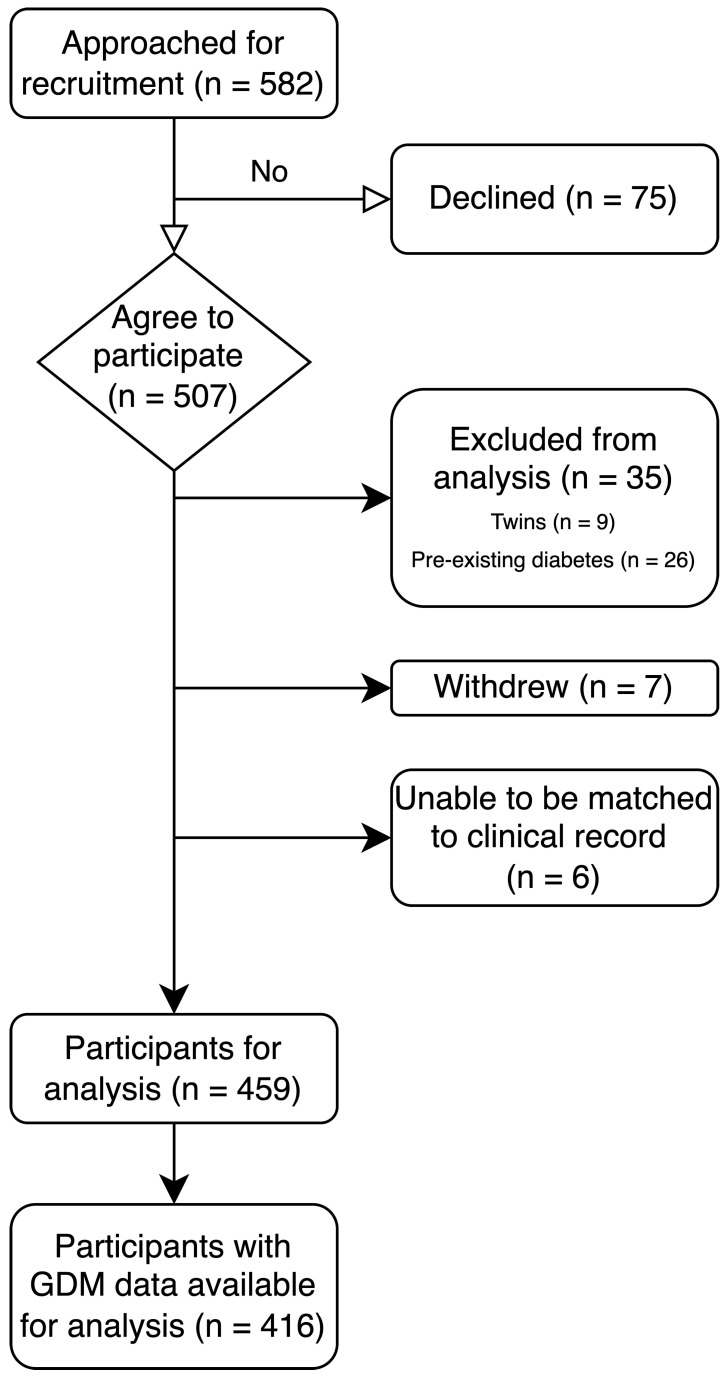
Participant recruitment.

**Table 1 nutrients-17-03500-t001:** Descriptive characteristics of participants.

Maternal		All Participants (*n* = 459)	Participants with GDM Data Available (*n* = 416)
	Age, median (range)	33 (21–43)	32 (21–43)
	20–24 years, *n* (%)	21 (4.5)	22 (5)
	25–34 years, *n* (%)	265 (58)	252 (61)
	35–39 years, *n* (%)	143 (31)	119 (29)
	40+ years, *n* (%)	30 (6.5)	21 (5)
	Parity, *n* (%)		
	Para 0	140 (30.5)	120 (29)
	Para 1	217 (47)	199 (48)
	Para 2	68 (15)	65 (16)
	Para 3	26 (6)	25 (6)
	Para 4+	8 (1.7)	7 (1.7)
	BMI kg/m^2^, mean (range)	25.3 (16.8–50.9)	26.5 (16.8–50.9)
	≤18.5	10 (2)	8 (2)
	18.5–25	211 (45)	195 (47)
	25–30	138 (30)	127 (31)
	≥30	100 (22)	86 (21)
	Smoking status, *n* (%)	9 (2)	9 (2)
	Reported alcohol use in pregnancy, *n* (%)	0 (0)	0 (0)
	Medical history		
	Hypertension, *n* (%)	7 (1.5)	6 (1.4)
	Diabetes	Excluded	Excluded
**Sociodemographic**			
	Self-reported ethnicity, *n* (%)		
	Middle Eastern	81 (18)	79 (19)
	South-East Asian	77 (17)	69 (16.5)
	South Asian	172 (37)	153 (37)
	White	90 (20)	79 (19)
	Other	39 (8.5)	36 (8.7)
	Country of birth, *n* (%)		
	Australia	143 (31)	131 (33)
	Overseas	316 (69)	285 (67)
	Number of people in household, median (range)	3 (1–12)	3 (1–12)
	Number of children in household, median (range)	1 (0–12)	1 (0–12)
	Education level completed, *n* (%)	Available for 457	Available for 414
	Incomplete high school	10 (2)	10 (2.4)
	Completed high school	38 (8.3)	36 (8.7)
	TAFE * certificate/diploma	82 (18)	74 (18)
	University/tertiary institute	317 (69)	285 (68)
	Other	4 (0.8)	3 (0.7)
	Prefer not to say/did not answer	6 (1.3)	6 (1.4)
	Household income bracket AUD, *n* (%)	Available for 454	Available for 411
	<50,000	23 (5)	22 (5)
	50–100,000	96 (21)	85 (20)
	100–200,000	153 (33)	139 (33)
	>200,000	75 (16.5)	64 (15)
	Don’t know/prefer not to say	107 (23.5)	101 (24)
	Missing	12	12
**Pregnancy**			
	Gestational age at recorded first visit in weeks, mean (range)	6.2 (5–13)	6.2 (5–13)
	Mode of conception, *n* (%)		
	Spontaneous	435 (95)	395 (94)
	IVF	20 (4.4)	17 (4)
	Ovulation induction	3 (0.6)	3 (0.7)
	Natural fertility services	1 (0.2)	1 (0.2)
	Edinburgh perinatal depression screening score, *n* (%)	Available for 455	Available for 412
	Score range 0–9	406 (89)	371 (90)
	Score range 10–12	31 (7)	27 (6.5)
	Score range ≥ 13 or response to Q10 on self-harm	18 (4)	14 (3.4)
	History of GDM, *n* (%)	63 (18) ^#^	54 (18) ^#^

^#^ of those with a previous pregnancy. * TAFE—Technical and Further Education, a vocational and training system in Australia with a focus on post-school vocational qualifications at certificate and diploma level.

**Table 2 nutrients-17-03500-t002:** Amount and intensity of physical activity reported.

Physical Activity Levels		*n* (%)
**By Category**	Sufficiently active	179 (39)
	Low active	206 (45)
	Inactive	73 (16)
**By Intensity**	Any walking	430 (94)
	Any moderate	85 (19)
	Any vigorous	74 (16)
	Any moderate/vigorous	133 (25)

**Table 3 nutrients-17-03500-t003:** Association between physical activity category, duration and type with diagnosis of gestational diabetes.

Physical Activity Levels		GDM (*n* = 104; 25%)	NO GDM (*n* = 312; 75%)	*p*
**By Category**	Inactive	15 (14%)	51 (16%)	0.624
	Low active	51 (49%)	136 (44%)	
	Sufficiently active	38 (37%)	125 (40%)	
**By Duration**	Total exercise (minutes per week)	113 (40–195)	120 (30–240)	0.671
	Walking (minutes per week)	80 (30–180)	90 (30–180)	0.915
**By Intensity**	Walking (any)	89 (86%)	278 (89%)	0.429
	Moderate (any)	15 (14%)	60 (19%)	0.338
	Vigorous (any)	10 (10%)	53 (17%)	0.097
	Moderate/vigorous (any)	21 (20%)	95 (30%)	0.058

**Table 4 nutrients-17-03500-t004:** Odds of developing GDM by physical activity levels.

Physical Activity Levels		Unadjusted		Adjusted *	
		OR (95% CI)	*p*	OR (95% CI)	*p*
**By Category**	Inactive	Ref.	Ref.	Ref.	Ref.
	Low active	1.28 (0.67–2.53)	0.470	1.07 (0.55–2.19)	0.839
	Sufficiently active	1.03 (0.53–2.09)	0.924	0.90 (0.45–1.87)	0.782
**By Intensity**	Any walking	0.73 (0.38–1.43)	0.336	0.63 (0.32–1.28)	0.185
	Any moderate	0.71 (0.37–1.28)	0.271	0.70 (0.36–1.30)	0.275
	Any vigorous	0.52 (0.24–1.02)	0.073	0.59 (0.27–1.20)	0.165
	Any moderate or vigorous	0.58 (0.33–0.97)	0.045	0.59 (0.33–1.01)	0.061

* Adjusted for age, BMI, ethnicity and multiparity.

**Table 5 nutrients-17-03500-t005:** Association between physical activity and oral glucose tolerance test results.

Physical Activity Level		Fasting Glucose	1 Hour Glucose	2 Hour Glucose
		Mean (SD)	Mean (SD)	Mean (SD)
**By Category**		*p* = 0.096	*p* = 0.125	*p* = 0.665
	Inactive	4.32 (0.52)	7.80 (2.16)	6.65 (1.79)
	Low active	4.47 (0.49)	8.21 (2.12)	6.86 (1.93)
	Sufficiently active	4.44 (0.41)	7.79 (1.85)	6.72 (1.77)
**By Intensity**	Any walking	*p* = 0.292	*p* = 0.559	*p* = 0.708
	Yes	4.44 (0.47)	7.96 (2.02)	6.76 (1.85)
	No	4.37 (0.44)	8.16 (2.14)	6.87 (1.81)
	Any moderate	*p* = 0.245	*p* = 0.369	*p* = 0.428
	Yes	4.38 (0.42)	7.78 (2.04)	6.62 (1.81)
	No	4.44 (0.47)	8.02 (2.03)	6.81 (1.86)
	Any vigorous	*p* = 0.409	*p* < 0.001	*p* = 0.316
	Yes	4.39 (0.38)	7.25 (1.68)	6.59 (1.52)
	No	4.44 (0.48)	8.11 (2.07)	6.81 (1.90)
	Any moderate or vigorous?	*p* = 0.187	*p* = 0.002	*p* = 0.109
	Yes	4.39 (0.38)	7.49 (1.87)	6.55 (1.66)
	No	4.45 (0.49)	8.17 (2.06)	6.86 (1.91)

## Data Availability

Data available on request due to restrictions as per the published protocol. Applications can be made to the Chief Principal Investigator and Corresponding author (D.P.).

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
