# Peer review of "Association Between Intensity of Physical Activity in Pregnancy and Gestational Diabetes in a Multi-Ethnic Population: Results from the PROMOTE Cohort Study"

_nutrients, 2025, doi:10.3390/nu17223500_

Round 1

Reviewer 1 Report

Comments and Suggestions for Authors

 The authors assess the influence of physical activity on the occurrence of GDM in 459 pregnant women <16GW recruited between 2022-2024, categorized as having sufficient, low, or inactive levels of physical activity. The authors compare the rates of GDM in each of these groups and observe a non-significant trend toward a reduced rate of GDM as the level of physical activity increases.

However, moderate to vigorous physical activity is associated with significantly lower 1-hour -OGTT glucose values. The authors suggest that this may explain the decrease in the rate of GDM associated with physical activity.

 I just have a few comments.

Major comments

The main problem of the paper concerns the estimation of the sample size, which is missing. What power does the study have to detect significant differences in the GDM rate associated with the categories of physical activity used? How did you estimate the sample size? With 459 participants, what power does the study have to detect differences >5% or >10% GDM? What population would be necessary to study to find that differences like those found in this study would be statistically significant? Please clarify/comment.

Minor comments

  1. In the abstract, include the recruitment period and the number, not just the percentage of each category
  2. - This is an association study between physical activity/diet/lifestyle and the occurrence of GDM in pregnant women with <16 weeks of gestation. It is essential to specify the time period to which the obtained data refer. The questionnaires on physical activity/diet/lifestyle completed by the participants, I assume, refer to the current moment (before 16 weeks of gestation), but does it refer to the time of the interview, that is, the first trimester of gestation, the period from when they found out they were pregnant until the current moment, or the pregestational situation? Please clarify.
  3. -- It is striking that 25% of the participants engaged in moderate-to-vigorous physical activity. Does this level of physical activity represent the population in this age range??? Please include some comments. What kind of sport do they practice? Swimming??
  4. - How are participants informed about the need to stop physical activity in the event of uterine contractions, abdominal pain, discharge… etc.? Please comment.-
  5. Moderate-to-vigorous physical activity is associated with a lower plasma glucose level 1 hour after OGTT, but it does not affect fasting glucose levels. - Considering that participants with GDM are identified more than 50% by FBG levels and only less than 20% by 1-hour OGTT values, how could they explain the reduction in the GDM rate?
  6. - It is known that physical activity mainly reduces endogenous glucose production, and FBG levels more than OGTT values. Can they make any comments on their findings?
  7. Do the authors consider adjusting for BMI class or categorizing their results by BMI? Is the level of physical activity inversely associated with BMI?

Author Response

Thank you very kindly for your time and expertise. Please see the attachment for our responses and summary of revisions.

With kindest regards,

Dr Ania Samarawickrama (Lucewicz) 

On behalf of the PROMOTE Cohort Study team

Reviewer 2 Report

Comments and Suggestions for Authors

The original article Association between physical activity intensity in pregnancy and gestational diabetes in a multi-ethnic population: Results from the PROMOTE Cohort Study assessed the relationship between the level of physical activity and the occurrence of gestational diabetes. Although a moderate level of physical activity is widely recommended, the protective effect on GDM is still debated, and research on this subject is important for primary prevention. The longitudinal design of the study provides higher evidence for the role of physical activity in GDM prevention, supporting the observations derived from cross-sectional studies. The introduction highlights the existing knowledge gap and motivates the study's purpose. The objectives of the study are clearly stated in the Introduction section – to evaluate the “association between participation in physical activity in pregnancy, of varying intensities and durations, and gestational diabetes” and to examine the associations between levels of physical activity, results of oral glucose tolerance tests (OGTT), and clinical and social characteristics of the participants.

The Materials and Methods section provides an in-depth description of the questionnaires used to evaluate the level of physical activity, the level of understanding of public health messages, and mental health disorders.

The presentation of the results is clear and provides sufficient descriptive information. In my opinion, subsection 3.1. “Participants” should be moved to the Materials and Methods section because it is related to the design of the study.  The association between the level and type of physical activity and the prevalence of GDM, and the results on the OGTT are clearly presented in both the text and the tables. However, I suggest explaining the results provided in parentheses in Table 5 ( are they standard deviations or standard errors?).

The discussion section addresses the results of the study and their integration into current scientific production. The lack of nutritional data represents an important limitation of the study, which I consider important to discuss. Using previously validated questionnaires mitigates another important limitation, which is the lack of objective quantification of the physical activity level.

The conclusions reflect the results of the study and emphasize that there should be further research on the role of physical activity in the prevention of GDM.

Author Response

(The authors gave the same response as above.)

Reviewer 3 Report

Comments and Suggestions for Authors

This is a well-executed cohort study examining the relationship between physical activity intensity during pregnancy and gestational diabetes in a multi-ethnic Australian population. The manuscript is well-structured and the data are thoughtfully analyzed. With minor revisions, it will make a valuable contribution to Nutrients.

1] Modify phrases such as “physical activity was associated with lower glucose levels” to “showed an association with lower glucose levels” to maintain observational neutrality.

2] Supplementary Tables S2–S4 are misnumbered in the text.

3] The manuscript should specify how missing data were managed (e.g., listwise deletion, imputation). Even a brief clarification in the Methods would strengthen transparency.

4] Several p-values (e.g., p = 0.058; p = 0.061) indicate a trend rather than statistical significance. Please rephrase conclusions accordingly.

5] Abstact could be more concise and balanced.

6] The manuscript would benefit from an additional figure to visually summarize key findings.

Author Response

(The authors gave the same response as above.)
